



# On the value of high density rain gauge observations for small Alpine headwater catchments

Anthony Michelon[1], Lionel Benoit[1], Harsh Beria[1], Natalie Ceperley[1], Bettina Schaefli[1,2]

[1] Institute of Earth Surface Dynamics (IDYST), Faculty of Geosciences and Environment, University of Lausanne, Lausanne, 1015, Switzerland
[2] Now at: Institute of Geography (GIUB), Faculty of Science, University of Berne, Switzerland

*Correspondence to*: Anthony Michelon (anthony.michelon@unil.ch)

**Abstract.**

Spatial rainfall patterns exert a key control on the catchment scale hydrologic response. Despite recent advances in radar-based rainfall sensing, rainfall observation remains a challenge particularly in mountain environments. This paper analyzes the importance of high-density rainfall observations for a 13.4 km$^2$ catchment located in the Swiss Alps where summer rainfall events were monitored during 3 months using a network of 12 low-cost, drop-counting rain gauges. We developed a data-based analysis framework to assess the importance of high-density rainfall observations to help constrain hydrologic processes. The framework is based on two steps, the identification of key hydro-meteorological metrics that explain the runoff coefficient and lag times (e.g. total event rainfall, center of mass of the precipitation field) and the identification of the optimal rain gauge network density to reliably reproduce the value of these metrics. The hydrological metrics are evaluated through correlation and regression analysis, resulting in the identification of three main drivers for the runoff coefficient and for runoff response lag times: the areal rainfall, the spatial asymmetry of the rainfall field and the antecedent rainfall over the three days preceding an event.

The relationships between these measures and the optimal network density gives insights into the importance of reliably observing the localisation of incoming rainfall. Even at the small spatial scale of this case study, the results show that an accurate representation of the rainfall field, with at least two rain gauges, is of prime importance to understand the hydrologic response. The largely data-based analysis framework developed here is readily transferable to other settings. Given the low cost of the deployed rainfall sensor network, the approach has potential for future detailed studies in to-date sparsely observed catchments. Future work could in particular also refine the presented analysis by improving the design of the rain gauge deployment to ensure a good representation of geomorphological units and of the relative distances to the stream network.

## 1 Introduction

Rainfall is known to be highly variable in space even at small scales, in particular in mountain areas (Henn et al., 2018; Tetzlaff and Uhlenbrook, 2005). Despite recent progress in the observation of spatial rainfall with the help of radar (Berne and Krajewski, 2013; Germann et al., 2006), it remains crucially difficult to observe and spatially interpolate (Foehn et al., 2018a).



Understanding the interrelation between spatial rainfall patterns and the hydrologic response has been of concern for many decades, ranging from a theoretical viewpoint (Shah et al., 1996; Singh, 1997; Woods and Sivapalan, 1999), to a rainfall-runoff model perspective (Obled et al., 1994), and extending to a hydrological process understanding perspective (Guastini et al., 2019; Zillgens et al., 2007). Even earlier work in this field focused on the model-based investigation of optimal rain gauge

density for reliable areal rainfall estimation (Bras and Rodriguez-Iturbe, 1976a) and runoff prediction (Bras and Rodriguez-Iturbe, 1976b; Tarboton et al., 1987). Chacon-Hurtado (2017) provides a recent review on rain gauge network optimisation.

There is a wide range of methods to analyze the hydrologic response as a function of spatial rainfall patterns. We can broadly distinguish between empirical methods that identify systematic response patterns by scrutinizing individual observed events (Blume et al., 2007) and model-based methods that try to identify systematic or theoretical relationships between rainfall and

the hydrologic response. In this latter category, we first of all find stochastic methods that describe the stochastic aspects of the hydrologic response as a function of the rainfall field properties. These approaches range from simplified stochastic models (Tarboton et al., 1987) to full space-time representations of rainfall forcing and streamflow generation (Mei et al., 2014; Pechlivanidis et al., 2017; Viglione et al., 2010; Woods and Sivapalan, 1999; Zoccatelli et al., 2015). These stochastic tools are developed to understand the relative importance of the two key components of the hydrologic response, i) the runoff

generation processes at the hillslope scale and ii) the routing mechanisms in the channel network. Such an assessment of the relative role of unchanneled-state and channeled-state processes (Rinaldo et al., 1991b; Rinaldo et al., 2006a) gives key insights into the relative role of runoff generation processes and the geomorphology of a catchment. This can also be achieved with virtual modelling experiments with hydrological models that explicitly account for geomorphological dispersion along the channel network.

An example is the work of Nicótina et al. (2008) who assessed the importance of well representing spatial rainfall variability for medium size catchments (a few hundreds to thousands km²) where saturation-excess overland flow dominates (rather than Hortonian flow). They conclude that for rainfall events with a spatial correlation length larger than the hillslope size, an exact representation of the spatial rainfall variability is not required to well represent the hydrologic response - provided that the mean area rainfall is preserved at each time step. They explain this result by the fact that if the total catchment-scale residence

time is controlled by the travel time within the hillslopes, large enough rainfall events sample all possible residence times, independent of the actual spatial rainfall configuration. Their findings were subsequently confirmed by the work of Volpi et al., (2012) amongst others, where a simplified modelling approach based on a geomorphological unit hydrograph was used. While the conclusions were similar, this study also added that spatial variability does not matter "when the integral scale of the excess-rainfall field is much smaller or much larger than the basin drainage area".

Similar results were obtained in numerous studies that assess the impact of undersampling or of coarse graining an observed rainfall field on the performance of streamflow simulations obtained with more or less complex process-based hydrologic models (Bardossy and Das, 2008; Moulin et al., 2009; Lobligeois et al., 2014; Shah et al., 1996; St-Hilaire et al., 2003; Stisen and Sandholt, 2010; Xu et al., 2013). A key result of all these  model-based studies is  the fact that the hydrologic response depends more on an accurate estimate of the mean areal rainfall than on the actual exact form of the rainfall field, requiring



high density rainfall observations (Obled et al., 1994). However, such studies typically use models that cannot explicitly assess the relative role of channeled versus unchanneled-state processes, which makes their results less generalizable. Furthermore, these studies often suffer from the general problem that conceptual hydrological models require recalibration if used with different input fields, which makes disentangling effects from rainfall versus parameters a cumbersome exercise (Bardossy and Das, 2008; Bell and Moore, 2000; Stisen and Sandholt, 2010).

This hypothesis that the area-average rainfall might play a more important role for the streamflow response than the actual spatial rainfall pattern remains to be tested in the field. In this paper, we propose therefore a purely data-based analysis of the importance of rain gauge density for the event-specific hydrologic response (Ross et al., 2019) of a small, high elevation Alpine headwater catchment (13.4 km$^2$) where the hydrologic processes have been intensely monitored since 2015. A small catchment has potential to disprove the assumption that for catchments smaller than a few tens of km$^2$, a single rain gauge is sufficient

for reliable runoff prediction, which stems from the majority of studies on the influence of rainfall patterns that typically analyze larger catchments.

Despite being located in Switzerland with a relatively high density rain gauge network and a well-developed radar network (Foehn et al., 2018b; Sideris et al., April 2014), none of the existing rainfall observation networks adequately observes spatial rainfall patterns in the area of interest (Benoit et al., 2018a; Benoit et al., 2018b). We therefore deployed a network with

commercially available, low cost, drop-counting rain gauges, which increases the interest of this case study for future hydrologic studies in similar settings.

Based on the high density rain gauge observations during one summer (approximately one rain gauge per km$^2$), we developed an analysis framework to answer two key questions:

i.    What is the effect of the spatial location of rainfall fields on the timing and amplitude of the hydrologic response?

ii.   What does the high density rain gauge network contribute to assess the main characteristics of the rainfall field?

This last question provides insight for the design of a future permanent rainfall observation network.

The framework to answer the above questions relies on the definition of appropriate metrics (see Section 3) to characterize the observed spatial rainfall patterns as well as the hydrologic response features. These include metrics classically used in process hydrology, such as the runoff coefficient (Blume et al., 2007) or antecedent rainfall (Ross et al., 2019) as well as metrics

generally used in modelling to characterize the water residence time in the channeled versus the unchanneled state.

## 2  Case Study

The area of interest is the Vallon de Nant, a 13.4 km² catchment located in the Western Swiss Alps (Figure 1). The elevation ranges from 1,200 masl. at the outlet of the Avançon de Nant river to 3,051 masl. (Grand Muveran), and have an average elevation of 1,975 masl. The catchment benefits from a protected status (Natural Reserve of the Muveran) since 1969 and is

of national importance for Switzerland in terms of biodiversity (Cherix and Vittoz, 2009). The Vallon de Nant has been intensively studied over the recent years, in disciplines ranging from  hydrogeology (Thornton et al., 2018), to pedology (Lane





et al., 2016; Rowley et al., 2018), to biogeochemical cycling (Grand et al., 2016), and even stream ecology (Horgby et al., 2019).

The Vallon de Nant belongs to the reverse side of the Morcles nappe, a structural geological unit that determines the
catchment's shape. The old Cretaceous and Tertiary layers are recognizable as a succession of thick, blocky lithologies overlooking and surrounding the valley. They lie on a substratum of flysch, i.e. softer rocks (schistose marls and sandstone benches), which explains the deepening and widening of the valley at its southern part (Badoux, 1991).

Figure 2 summarizes the dominant hydrological units of the Vallon de Nant. The western side is mainly characterized by grassy slopes, with deep soils and a relatively high water storage capacity as revealed by gauging along the stream during the
late summer and autumn yearly streamflow recession period (Horgby, 2019). The northern part of these western slopes shows a less dense drainage network than the rest of the catchment (Figure 1), explained by steeper slopes, a large hydraulic conductivity or locally deeper soils.

The eastern side of the catchment is characterized by steep and rocky slopes that react quickly to rain events due to shallow soil that drains quickly. At the foot of the rock walls, large alluvial cones and scree extend down to the river. The bottom of
the valley is mainly composed of fine alluvial deposits with a large water storage capacity. In the upper part of the valley, the Glacier des Martinets (area less than 1 km$^2$) is now confined to a small area shaded by the Dents de Morcles. The water flow paths of rainfall inputs over this upper part of the catchment, composed of moraines and permafrost remain unclear and have not been investigated so far.

The Avançon de Nant river shows a typical snow dominated streamflow regime marked by a high flow period during spring
and early summer when the snowpack accumulated during the winter melts (see Figure S2 in the Supplementary Material). Its length reach 6 km in early summer, while during autumn and winter low flow the river may start to flow as low as 1480 masl. (close to the gauge No. 5 on the Figure 1), reducing the instream flow distance to the outlet to 2.95 km. The actual extent of the stream network is based on observations during dry and wet periods and its exact path was calculated using a digital elevation model.

The streamflow at the outlet is observed via river height measurements using a sonar above the middle point of a trapezoidal shaped weir (see picture in the Supplementary Material S51). It averages water height every 1 minute continuously since September 2015. The height is then converted into streamflow using a rating curve based on 23 salt discharge measurements (Ceperley et al., 2018). The annual average streamflow is about 0.67 m$^3$.s$^{-1}$ (from January 1$^{st}$ 2016 to December 31$^{st}$ 2018); average annual water temperature is 5.0°C, ranging from a frozen river during some days in winter to an average temperature
of 8.5°C during summer (from July 1$^{st}$ to August 31$^{st}$, 2017). The maximum streamflow measured at this station is 12.7 m$^3$.s$^{-1}$ during an intense rainfall event (August 6$^{th}$, 2018).

Meteorological variables are monitored at three locations (Michelon et al., 2017) along a north/south transect (at 1253 masl., 1530 masl. and 2136 masl.) since September 2016. The average air temperature at mean elevation estimated from these stations equals 3.1 °C in 2017.



Of particular note, we do not use the Swiss weather radar network (Sideris et al., April 2014) since the Vallon de Nant catchment is in the shadow of existing radars (Foehn et al., 2018b). The weather radar might see here at best rainfalls above 2800 masl. (Marco Gabella personal communication, February 27th 2019).

## 3    Method

The methodological framework developed to analyze the rainfall events, the hydrological response and ultimately the optimal
rain gauge density can be summarized as follows: i) define the appropriate metrics to describe the rainfall fields and hydrological response, ii) understand relationships between these metrics through correlation analysis, iii) identify main drivers (i.e. the corresponding metrics) through regression analysis, and iv) use the gained insights to optimize the rain gauge network based on selected metrics.

### 3.1    Instruments

A network of 12 Pluvimate drop-counting rain gauges (www.driptych.com) was distributed across the Vallon de Nant catchment from July 1st to September 23rd 2018 in order to capture summer rainfall events (Figure 1). A similar deployment during the cold season would not be possible due to snowfall at all elevations throughout the winter. The sites were selected to represent the distribution of slope orientations and elevation, but also to meet constraints of accessibility and disturbance risk (livestock, hikers). The distance between measurement locations within the network ranges from 350 m to 1,550 m (630
m on average), and the greatest distance from any point in the basin to a rain gauge is 1,670 m.

The gauges are low-cost (around 600 USD each),  consisting of a tube (11 cm of diameter, 40 cm of length) mounted to an aluminum funnel (Figure 3). The collected rainwater is concentrated to a nozzle that creates a drop of water of calibrated size (0.125 mL), which then falls on the impact-sensitive surface of the sensor, 30 cm below. The datalogger counts and records the number of drops over a time set up to 2 minutes. In the field, the devices are set up vertically, attached to a wooden stick.
The funnel aperture is between 0.8 and 1.2 m above the ground.

The Pluvimates were set-up to count drops over an interval of 2 minutes, with an accuracy of 0.3 mm/h. Benoit et al. (2018a) experimentally evaluated the device uncertainty to 5 % with rainfall intensities under 20 mm/h. Saturation of the nozzle is expected at higher intensities. Given some of the rainfall intensities measured during the period of observation (up to 140 mm/h), we extended the experimental tests to intensities up to 150 mm/h (see Appendix A).
To prevent clogging, steel sponges were disposed at the entrance of each Pluvimate (see Figure 3). This appeared to have caused i) a dampening effect on low rainfall intensities as it delayed slightly the beginning of events (lower than 1 mm/h) and ii) created drops remaining after the end of an event. The data are not corrected for these effects.

Some additional artefacts were recorded, probably generated by strong winds creating resonance, which noised the data. These supposed windy periods have been manually removed from the data.





## 3.2    Rainfall events identification

Before further analysis, the rainfall amounts measured by each station were interpolated to a 10 by 10 m grid at a 2 min time step using the Thiessen polygons method. This extremely simple spatial interpolation method is justified for the high density rain gauge network at hand, with a rain gauge density of 0.5 to 1 station per square kilometer (see, e.g., Syed et al., 2003).

When part of the measurement network was not functional, the spatial imprint of the functional stations was adapted to exclude non-operational parts. This partial network configuration occurred mainly at the beginning and at the end of the observation period, when the number of operational stations was gradually increasing to and decreasing from the full network of 12 stations, and during the core observation period when the recordings of some devices were manually discarded due to wind effects (see Section 3.1).

Using the interpolated rainfall field, rainfall events were distinguished by periods of at least 90 minutes with no rainfall. This inter-event duration was selected based on the observed delay between rainfall onset and streamflow response for the large event recorded on August 23$^{rd}$ at 4:26 pm (detailed in Supplementary Material S36); the streamflow reaction to the first half-hour of this rainfall event was caused only by rainfall in the southern half of the catchment (stations 8 to 12). Ninety minutes was retained to maximize the chances of observing a distinct streamflow reaction for two distinct consecutive events. In addition, events with a total amount of rainfall under 1 mm are discarded for this study.

## 3.3    Streamflow response identification

The beginning and the end of each streamflow event were identified manually using a data visualization tool (developed in Mathworks MatLab 2017a, see Figure 4 and Figure 5). This choice of a visual expertise was made based on the observation that automatic identification of streamflow events would require almost a case-by-case filtering and parametrization and thus would not be generalizable. This is partly related to a potentially high signal-to-noise ratio for river stage recordings during sediment transport events, a phenomenon potentially very important after a strong streamflow variation. The result of this visual identification for each streamflow event is in the Supplementary Material (Figures S3 to S50).

The beginning and the end of the streamflow response determine the initial and final baseflow; the streamflow volume above the line connecting these two points is considered here as fast runoff. It is noteworthy that we do not use peak streamflow to characterize streamflow events, for two reasons: i) given the small size of the catchment and the complex temporal distribution of rain intensities, the streamflow response has rarely a single, well identifiable peak; ii) peak streamflow identification is further complicated due to noise in the stage recordings.

## 3.4    Rainfall and hydrological response characterization

To further analyse the relationship between the spatial distribution of rainfall and of the hydrological response, we use a set of metrics, described in detail hereafter and summarized in Table 1. For a detailed discussion of rainfall-runoff response metrics, see the paper of Ross et al. (2019).





### 3.4.1 Spatial rainfall patterns

To investigate the relationship between dominant spatial rainfall patterns and streamflow response, the catchment area that spreads latitudinally is split into two parts of equal area by a west-east line (see Figure 1), delimiting an area close to the outlet in the lower (northern) part, and an area farther away in the upper (southern) part. The interpolated amounts of rainfall received by the upper and lower parts of the catchment (respectively $P_{UPPER}$ and $P_{LOWER}$) are compared and normalized by the total amount of rainfall to create an index of spatial rainfall asymmetry $I_{ASYM}$:

$$I_{ASYM} = \frac{P_{UPPER} - P_{LOWER}}{(P_{UPPER} + P_{LOWER})},$$ (1)

The rainfall is equally distributed between the upper and lower parts when $I_{ASYM} = 0$. The extreme values -1 and 1 describe rainfall focused exclusively within the lower or upper part of the catchment, respectively. We consider a rainfall as asymmetric when at least 2 times more rainfall is precipitated over one part of the catchment than over the other; the corresponding $I_{ASYM}$ threshold values are -0.33 and +0.33 (respectively 2 times more rainfall precipitated over the lower or upper part than over the other part).

To further analyse the relationship between the spatial distribution of rainfall and the streamflow response, we characterize the geomorphological distance of incoming rainfall from the outlet, assuming that this distance should reflect to some degree the time of reaction of the catchment: following the terminology of Rinaldo et al. (2006b), transport at the basin scale can be analysed in terms of travel in the unchannelled state (i.e. in the hillslopes) and travel in the channelled state (stream network). Following this approach to quantify the effect of geomorphological dispersion (Rinaldo et al., 1991a) on the streamflow response, we estimate the centre of mass of each rainfall event using the weighted average distance to the stream network. The stream network to its maximal extent (Figure 1) is determined manually once by setting the uppermost points of the catchment where streamflow has been observed in the field, and calculating its flow path based on the 2 x 2 m digital elevation model (DEM) swissAlti3D (2012). The flow distance of each cell to the stream network $d_{HILLS}$ is calculated once using the flow path deduced from the same DEM. The weight given to each $d_{HILLS}$ is the amount of precipitation cumulated over the event $P_{EVENT}$ at each point. The distance of the rainfall centre of mass on the hillslopes $D_{HILLS}$ is given by the average value:

$$D_{HILLS} = \frac{\sum_i \sum_j P_{EVENT}(i,j) d_{HILLS}(i,j)}{\sum_i \sum_j P_{EVENT}(i,j)},$$ (2)

where $i$ and $j$ are the line and column index of each cell within the grid.

Similarly, we compute the weighted average distance for rainfall between its point of introduction into the stream network and the outlet. For each cell of the stream network, the distance to the outlet is calculated once based on the 2 x 2 m DEM. The distance of each cell to the outlet $d_{STREAM}$ is weighted by the amount of rainfall potentially arriving at each cell of the stream network during an event $C_{EVENT}$. The distance of the rainfall centre of mass within the stream network $D_{STREAM}$ is given by the average value:

$$D_{stream} = \frac{\sum_i \sum_j C_{EVENT}(i,j) d_{STREAM}(i,j)}{\sum_i \sum_j C_{EVENT}(i,j)}.$$ (3)



The $D_{\mathrm{HILLS}}$ metric give an estimate of the average distance that incoming rainfall has to travel on the hillslopes before reaching the stream network, and $D_{\mathrm{STREAM}}$ the average distance for the water particle entering the stream network to reach the outlet.

### 3.4.2 Hydrological response metrics

Initial wetness conditions have been largely studied in the context of streamflow generation and are known to be the major parameter explaining the difference in the dynamics of the hydrological response of different catchments (Penna et al., 2011; Rodriguez-Blanco et al., 2012), in particular through the creation of runoff thresholds (Zehe et al., 2005; Tromp-van Meerveld and McDonnell, 2006). Here, initial wetness conditions are quantified in terms of antecedent rainfall, i.e. using the cumulative rainfall (in mm) that occurred during a period from 1 to 5 days before an observed rainfall event.

The runoff coefficient (RC) relating the fast streamflow component to the rainfall magnitude is computed at the catchment scale on an event basis from the ratio of the fast runoff (see section 3.3) and the interpolated cumulated rainfall per event. Lag times are usually used to characterize hydrological systems (Cuevas et al., 2019). Here, we estimate lag times for each rainfall event: $P_{\mathrm{START}}\_Q_{\mathrm{START}}$ is the time elapsed between the start of a rainfall event and the start of the river reaction (identified visually, see Section 3.3).

### 3.5 Regression analysis

We use pure quadratic regression to relate rainfall pattern characteristics and initial wetness conditions to the runoff coefficient. The list of the tested predictors is summarized in the Table 1. Significance of each regression is first tested using a p-value ($< 0.05$) and then compared using the Akaike Information Criterion (AIC, noted here $I_{\mathrm{AIC}}$) to evaluate the best model (Akaike, 1974):

$$ I_{AIC} = n \ln\left(\frac{S_{RSS}}{n}\right) + 2k + C \;, \tag{4} $$

where $n$ is the number of events, $k$ the number of coefficients, $S_{\mathrm{RSS}}$ the residual sum of squares and $C$ a constant that can be ignored when comparing different models based on the same data set. We also compare the different models using a corrected version of the AIC (AICc, noted here $I_{\mathrm{AICc}}$) for small sample sizes (Burnham et al., 2011):

$$ I_{AICc} = I_{AIC} + \frac{2k(k+1)}{n-k-1} \tag{5} $$

For both AIC and AICc, the best model is the one having the lowest score.

### 3.6 Measurement network configuration analysis

Assuming that the actual rainfall measurement network is sufficient to capture the full spatial distribution of rainfall in the studied catchment, we assess the ability of partial networks to reproduce the key metrics of the rainfall field for hydrologic process research. These metrics are not known *a priori* and are partly identified within the present analysis framework based

on correlation and regression analysis. The aim is twofold: i) identifying the best configuration for a future permanent





observation network and ii) evaluate the added value of additional rain gauges in a partial network with respect to the identified key metrics (see Section 4.3 and 4.4).

The quality of a partial network configuration is evaluated comparing the value by event obtained with the partial network to the reference value obtained with the full network setup. We evaluate all the possible combinations of partial networks

composed of less than 12 stations, i.e. 4094 possibilities. Each configuration is evaluated by computing the root mean square error (RMSE):

$$\text{RSME} \coloneqq \sum_{\forall t} X_k(t) - X_{\text{ref}}(t), \tag{6}$$

where $X_k$ is the selected rainfall metric (e.g. rainfall amount) at time step $t$ corresponding to the $k$-th network configuration, and $X_{\text{ref}}$ the respective value obtained reference network set-up.

The best network for each number of stations is the one with the lowest RMSE. A sensitivity analysis is completed by removing from 1 to 3 rainfall events to the 23 events dataset, yielding 2047 datasets evaluated for each partial network configuration. The most frequent network configuration validates the robustness of the result.

## 4   Results

### 4.1   Observed rainfall events and rainfall patterns

The available 3-month measurements window between July 1$^{\text{st}}$ and September 23$^{\text{th}}$ 2018 captured 48 rain events (detailed in the Supplementary Material, Table S1) for a total rainfall amount of 345.1 mm. The areal rainfall amounts per event range from 1 mm to 42.4 mm for events lasting between 32 minutes and 10.5 hours. Despite the sequential deployment of the 12 rain gauges and other technical issues (see section 3.1), the rainfall events were all measured by at least 7 stations; 36 out of 48 events were recorded by at least 10 stations and 23 events were recorded by the full setup of 12 stations. Details for all recorded

rainfall events and the corresponding streamflow recordings are visualized in summary plots as illustrated in Figure 4 and Figure 5 (all events are presented in the Supplementary Material). Most events show a relatively homogeneous spatial distribution of rainfall (Figure 5). A few events show very particular spatial rainfall patterns or hydrologic response patterns, as discussed further below.

One such event occurred on July 24$^{\text{th}}$ at 6:32 PM (Figure 4). The map shows a heterogeneous distribution of rainfall, centered

close to the outlet in the northern part of the catchment, over 6 out of the 12 stations. One of the rain gauges recorded up to 35.3 mm of rainfall, whereas 1.8 km upstream, half of the stations (on the southern and western parts of the catchment) did not record any rainfall. The interpolated amount of rainfall over the basin was 10.6 mm, and a fast runoff volume of 31.3 mm was measured (Table S1), resulting in a runoff coefficient of 3.0 that remains difficult to explain. One possible explanation is that important rainfall amounts fell on the north-eastern parts, over steep slopes that are difficult to access and where no rain

gauges were located.



### 4.2    Hydrologic response

For 6 of the 48 rainfall events (13 days in total), the water stage sensor was disturbed by the proximity of a rock (see picture in the Supplementary Material S52), resulting in missing streamflow data. For the remaining 42 rainfall events, a streamflow response was observed for 15 of them (see summary of the metrics for all events in the Supplementary Material, Table S1).

The role of initial wetness conditions is discussed qualitatively by comparing a pair of rainfall events with very similar spatial patterns and amounts, presented in Figure 5. For the first event, the measured rainfall ranges from 6.2 mm to 11.8 mm, corresponding to 8.5 mm of rainfall over the catchment in 2 h 38 min. For the second event, the rainfall ranged between 5.4 mm and 11.4 mm, corresponding to 8.4 mm over the catchment during 1 h 14 min. Despite the similar total amount of rainfall and event duration (during the first event 76 % of the total rain happened for a duration similar to the second event), the first

event shows a fast runoff volume of 7.4 mm, whereas for the second event the streamflow response is almost invisible. This difference can be explained by the initial wetness conditions, with 24.1 mm of rainfall during the 12 hours preceding the first event, compared to no rainfall during the preceding 3 days for the second event.

### 4.3    Correlation between rainfall patterns and streamflow response metrics

The possible links between the rainfall pattern and streamflow response metrics are investigated through a linear correlation

analysis (Figure 6) for those metrics of Table 1 that can be assumed to have a physical link.

As shown in Figure 6A, some events have an asymmetric spatial distribution of rainfalls (8 out of 48 events) and thus plot outside of the 1/2 and 2/1 lines (2 times more rainfall in one part of the catchment than in the other). On the hillslope centroid vs. river centroid distances (Figure 6B), the asymmetric rainfall events show a markedly distinct behaviour, which suggests that the distance metrics, jointly, might be useful to discriminate symmetric from asymmetric events, but that with respect to

$I_{\mathrm{ASYM}}$, these metrics might be redundant. Interestingly, the asymmetry mainly concerns low total rainfall amounts, with 7 out of 8 events concerned receiving below 5 mm (Figure 6A).

As visible on the plot of fast streamflow against rainfall total (Figure 6C), there is a clear threshold about 5 mm for runoff formation below which no streamflow reaction can be detected. This threshold reflects the role of initial wetness in fast runoff generation processes (see section 4.2 and 4.4) but is partly influenced by the noise of river stage recordings.

The outlier flagged in some subplots of Figure 6 represents the July 24th event already discussed in Section 4.1 with an unreasonably high runoff coefficient, and similar reasons of rainfall field underestimation can be invoked to explain the outliers having a runoff coefficient higher than 1 in the plot of the runoff coefficients against rainfall totals (Figure 6C).

The plots of initial baseflow against antecedent rainfall show (Figure 6H to K) that the initial baseflow before an event cannot be used as a proxy for initial wetness conditions, as it is uncorrelated ($R^2 \leq 0.1$) to antecedent rainfalls. It is clearly visible on

these plots that the relation between antecedent rainfall and initial baseflow depends on the period of the summer (points in Figure 6H to K are grouped by month); we furthermore see a shift from high to low baseflow values across time. The reason



for this is that the period of study (summer) corresponds to a long-lasting recession occurring yearly after the maximum streamflow generated by the snowmelt around mid-May (see Supplementary Material S2).

Figure 6E and F show $P_{START}\_Q_{START}$ lag time against the hillslope centroid and river centroid distances, but there is no

significant correlation between these metrics (respectively R²=0.11 and R²=0.08). The hillslope centroid distance could furthermore be assumed to have a link with the RC since longer distances to the stream network could potentially offer more intermediate storage opportunities (and thus reduce the RC). However, no particular pattern can be seen in Figure 6G.

### 4.4    Hydrologic driver analysis

Based on the insights obtained from the correlation analysis, different regression models are tested to explain the runoff

coefficient RC (summarized in Table 2) and lag time (see Table S2 in Supplementary Material). Given the strong correlation between the two geomorphic distances ($D_{HILLS}$ and $D_{STREAM}$), we included only the geomorphic distance in the hillslope, $D_{HILLS}$, in the analysis. The analysis is based on 14 events (removing the outlier with RC = 3.0 from the 15 events with an identified streamflow reaction), accordingly, we include the corrected AIC score, AICc for model ranking (see Section 3.5). These two model metrics show a strong correlation but rank some models differently (Table 2).

The AIC and AICc scores do not yield the same model ranking for RC (Table 2): a two predictors model (cumulated rainfall over the last 3 days and $I_{ASYM}$) is the best according to the AIC, but a single predictor model (cumulated rainfall over the last 3 days) gets the best score using the AICc. These models are the only ones having an acceptable p-value (<0.05), and therefore to be statistically acceptable.

For the $P_{START}\_Q_{START}$ lag times (see Table S2 in the Supplementary Material), no model has a p-value < 0.05. The ranking in

terms of AIC or AICc is however very similar to the ranking for RC, with the best ranked model with AIC being a two-predictor model with cumulated rainfall over the last 3 days and $I_{ASYM}$; with AICc, the single predictor model with cumulated rainfall over the last 3 days is first ranked. Furthermore, all models including $I_{ASYM}$ outperform the corresponding models including the geomorphologic distance $D_{HILLS}$. Based on these results, we keep the two metrics $I_{ASYM}$ and cumulated rainfall over the last 3 days as key metrics for the network density analysis.

### 4.5    Rain gauge network analysis

During the observation period, 23 out of 48 events were captured by the full network of 12 stations, measuring a total amount of rainfall of 120.7 mm. Based on the hydrologic driver analysis, we retain the rainfall amount, $I_{ASYM}$, and RC metrics for the network configuration analysis.

Figure 7 shows the best network configurations for 1 to 5 stations and the corresponding RMSE for the total rainfall per event,

but also based on the $I_{ASYM}$ and RC values per event.

For a 1-station network, the rainfall amount is better estimated when the station is located in the middle of the catchment, while a 2-station network improves substantially the RMSE by arranging the measuring points between the northern and southern





parts. Additional stations still improve the RMSE, although in a lesser extent. With a 4-station and 5-station network, the stations tend to align along a north-south transect.

To reproduce the RC, the arrangement of the stations and RMSE improvement tend to be the same from a 1-station to a 3-station network; then the spatial arrangement becomes totally different, preferring stations in the middle of the catchment. The behaviour cannot be explained except by the small size of the dataset (6 events) the network optimization is based on for this parameter.

As the $I_{ASYM}$ cannot be calculated from a single point, the 1-station network is used as reference network, comparing $I_{ASYM}$
values obtained from a complete network, with null $I_{ASYM}$ values of rainfall amounts evenly distributed between the upper and lower parts of the basin. The RMSE is therefore greatly improved by adding a station, by sampling precipitation in both parts of the basin. The enhancements from additional stations correspond to a better estimation of the rainfall amounts in the respective upper and lower parts of the basin, explaining the grouping of the stations into two groups.

Considering the small dataset underlying this analysis (23 events), the robustness of the best networks of Figure 7 (for the
rainfall amount and $I_{ASYM}$) is evaluated by re-computing the optimal network if between 1 and 3 events are removed from the error computation. This sensitivity analysis is not performed for the RC as 6 events only gather a full network setup and a river reaction. Figure 8 shows how frequent a given configuration is identified as being the optimal solution for networks composed of 1 to 3 stations and clearly confirms the optimal solutions found previously.

## 5  Discussion

### 5.1  Spatial heterogeneity of rainfall

The first metric used to evaluate the spatial distribution of rainfall splits the catchment into two parts, and averages rainfall measures into two values. Among the records showing a rainfall asymmetry, 7 out of the 8 events are too small to cause a detectable streamflow reaction (see Figure 6B), but one (see 4.1) does create a reaction although it only rains over half of the 12 rain gauge stations. However, our regression analysis based on 14 out of 48 rainfall events over a summer suggests that for
rainfall events that create a streamflow reaction, the spatial distribution might not play a rule for the explanation of the RC. The centroid distance metrics were designed to complete the rough asymmetry metric to locate the spatial rainfall centre of mass. However, the relationship between the rainfall centroid metrics and other metrics remains poor, with only a small correlation with the lag time ($R^2 < 0.13$), and no correlation with RC. The value of these spatial metrics for hydrologic response characterisation could be investigated in future studies considering the following: i) the stream network used for this study
corresponds to its maximal extent, which is not necessarily the extent observed at the beginning of an event, or even at the end of a small event; the exact dynamics of the stream network is not available for this study. It could be characterized in future studies e.g. with low cost temperature monitoring devices. ii) This network extent impacts both rainfall distance metrics: the hillslope centroid distance by potentially underestimating the distance between each grid cell and a stream; and the stream centroid distance by overestimating its extent and thus the distance to the outlet. This last metric is also sensitive to the rainfall





interpolation method used and the relative location of the rain gauges and how they are distributed in space: in the calculation of the rainfall centroid metrics, the weight given to a grid cell corresponds to the interpolated rainfall for this grid cell and thus depends on the relative position of the rain gauges. Figure 9A shows the distribution of distances between grid cells and the stream network for the spatial imprint of each rain gauge. While for many rain gauges the distribution of distances is similar, with a peak for small distances and a long tail, we clearly see that some rain gauges (stations 1, 2, 3 and 4) have a considerably

different distance distribution. The rain gauges 2, 7 and 12 are located in areas that do not show any superficial drainage network. As a result, the rain gauge spatial distribution is not optimal in terms of estimating the rainfall average distance to the stream network.

Going one step further with the network design, the spatial imprints could also be adjusted to the nature of the hydrological units (Figure 2). As shown in Figure 2, few rain gauges are represented predominately by a single unit. This configuration

should improve the correlation between the rainfall centroid metrics and both the lag time and runoff coefficient when capturing localized events. However, the increase of the number of criteria to be satisfied for the rain gauge locations (accessibility, average distance to the stream network, representativity for all hydrological units) might make the design a complex task.

### 5.2  Temporal variability of the lag time

It can generally be expected that rainfall events that are located farther away from the stream network or farther upstream along the stream network will show a longer lag time before causing a streamflow reaction at the outlet. Our dataset only shows a very weak positive trend in this sense (Figure 6E, F). The scattering and poor correlation might be explained by the rain gauge network disposition as discussed previously (Section 5.1). An additional explanation might come from the fact that we compare a metric over an entire event (rainfall centre of mass over the entire event duration) to the hydrological response

lag, which does a priori not depend on the spatial structure of the entire rainfall event. We could correct for this effect by considering rainfall spatial structure only over its starting period. Since the length of this starting period (and how it is related to intensity) is unclear at this stage, this would however introduce a source of analysis inhomogeneity. Another option is to assume that the lag time only starts after reaching the identified runoff threshold of 5 mm. (the total amount of rainfall that is at least necessary to observe a streamflow reaction). Another questionable point is the stream network used for the computation,

discussed hereafter in 5.3.

### 5.3  Rain gauge network density

The presented metrics showed the importance and potential of a high density rain gauge network to capture rain events and investigate the dynamics of the hydrologic response. The presented rain gauge network analysis can then be used as a preliminary investigation to implement a permanent network, composed of fewer stations. The reliability of the study is directly

dependent on the number of observed rainfall events, i.e. on deployment duration of the rain gauge network. Despite the small size of the catchment, there could potentially be storms that are not or only partially seen by the rain gauge network.




This possibility of missing localized events is given by the event of July 24[th] (see section 4.1), which was considerably underestimated despite of the high density of the deployed network (1 station for 0.9 km² on average, maximal distance of 1,670 m from a point to a rain gauge). The best partial networks composed of 1, 2 or 3 stations (see Section 4.5) give for this

extremely localized event a total amount of rainfall respectively 12.0 mm, 9.4 mm and 9.2 mm, not far from the 10.6 mm measured with the full network, but these partial networks were trained on the dataset containing the particular event.

With only one station, we can expect to totally miss an event, whereas a 2-station network design measuring at least the upper and lower part of the catchment would i) capture most of the events and ii) give a first estimation of the rainfall spatial distribution. It is noteworthy that beyond 3 stations the interest of an additional station gets weaker.

Finally, it is interesting to notice that the best network configurations (Figure 7) tend to design a linear north-south setup, an organization that can be explained by i) the shape of the catchment that also spreads longitudinally or ii) a general tendency for rainfalls to move longitudinally, emphasizing the importance, for this case study, to capture spatial configuration of rainfalls over a north-south transect rather than over a west-east transect and iii) the general increasing trend of elevation along this transect.

**6    Conclusion**

Our analysis of the role of rainfall patterns for the streamflow response is one of the first data-based studies at a comparable small scale in an Alpine environment. The detailed analysis of 48 events from one summer suggests that spatial rainfall patterns might indeed play a role even for such a small catchment. The novelties of the study include the use of a low-cost raindrop counting rain gauge network and the framework to analyse the rainfall-runoff response. The main conclusions from our

analysis are:

- A high density rain gauge observation network is a major asset to identify critical areas that are influenced by local rainfall forcing, and give an estimation of the rainfall amount errors made by a partial network.
- A detailed analysis of the hydrological response as a function of rainfall patterns and geomorphology requires a rain gauge network specifically designed for this purpose.

- Such a network should in particular take into account the spatial distribution of distances to the stream network since for small catchments these might vary considerably in space, depending on geological and geomorphological features. Such an analysis should account for the potential seasonal expansion of the stream network.




*Data availability.* Rainfall and streamflow data used for this paper, and the MatLab code written to visualize the data are available on Zenodo (Michelon et al., 2020).

*Author contributions.* AM and BS conceived the ideas and designed methodology; AM, LB and HB collected the rainfall data; AM and LB analyzed the data; AM and BS led the writing of the manuscript. All authors contributed critically to the drafts
and gave final approval for publication.

*Competing interests.* Author BS is a member of the editorial board of the journal, but otherwise there are no competing interests present that the authors are aware of.

*Acknowledgements.* The work of the authors is funded by the Swiss National Science Foundation (SNSF), grant number PP00P2\_157611.





## Appendix A: Drop-counting rain gauge calibration and data correction

Technical characteristics of the Pluvimate drop-counting rain gauges (see Section 3.1) are detailed in Benoit et al. (2018a); for

this study we extended the experimental tests to intensities up to 150 mm/h. It appears that for intensities up to 20 mm/h (99.88 % of the measured 2-min intensities during the 2018 observation period, see Figure A1) the linear relationship between drop count and rain intensity gives a good estimate (uncertainty below 5 %); beyond 20 mm/h the linear relationship underestimates the rainfall intensities, to reach 10 % of error at 60 mm/h and 15 % at 150 mm/h (Figure A1). For this study, rainfall intensities over 20 mm/h are corrected using a polynomial law based on the experimental measures. Thus the original amount of rainfall

measured through the observation period goes from 335.1 mm to 345.1 mm (Figure A2).

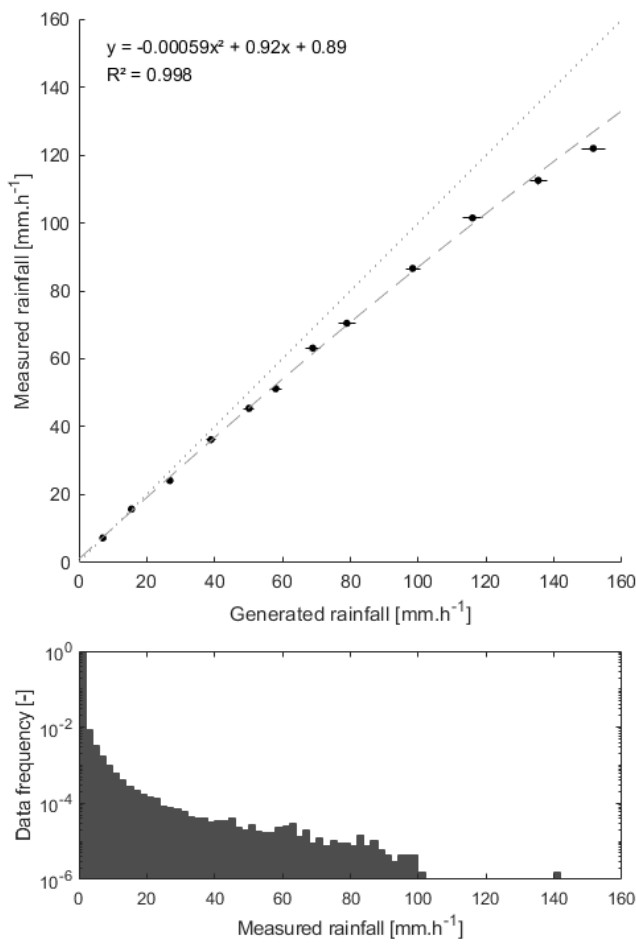

**Figure A1. Calibration curve (on top) of the Pluvimate rain gauges based on experimental measures with controlled rainfall input, and (at the bottom) the data frequency measured in situ.**





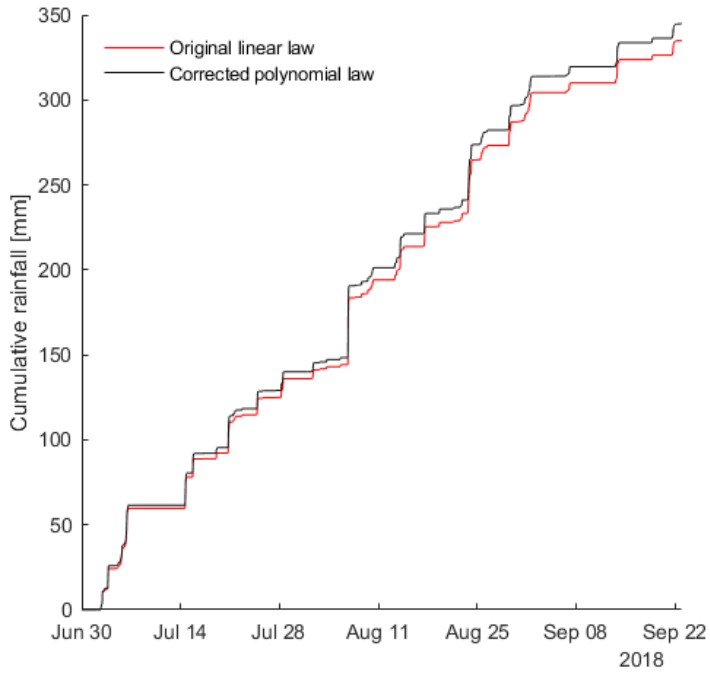


**Figure A2. Impact of rainfall data correction on the cumulative rainfall.**



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

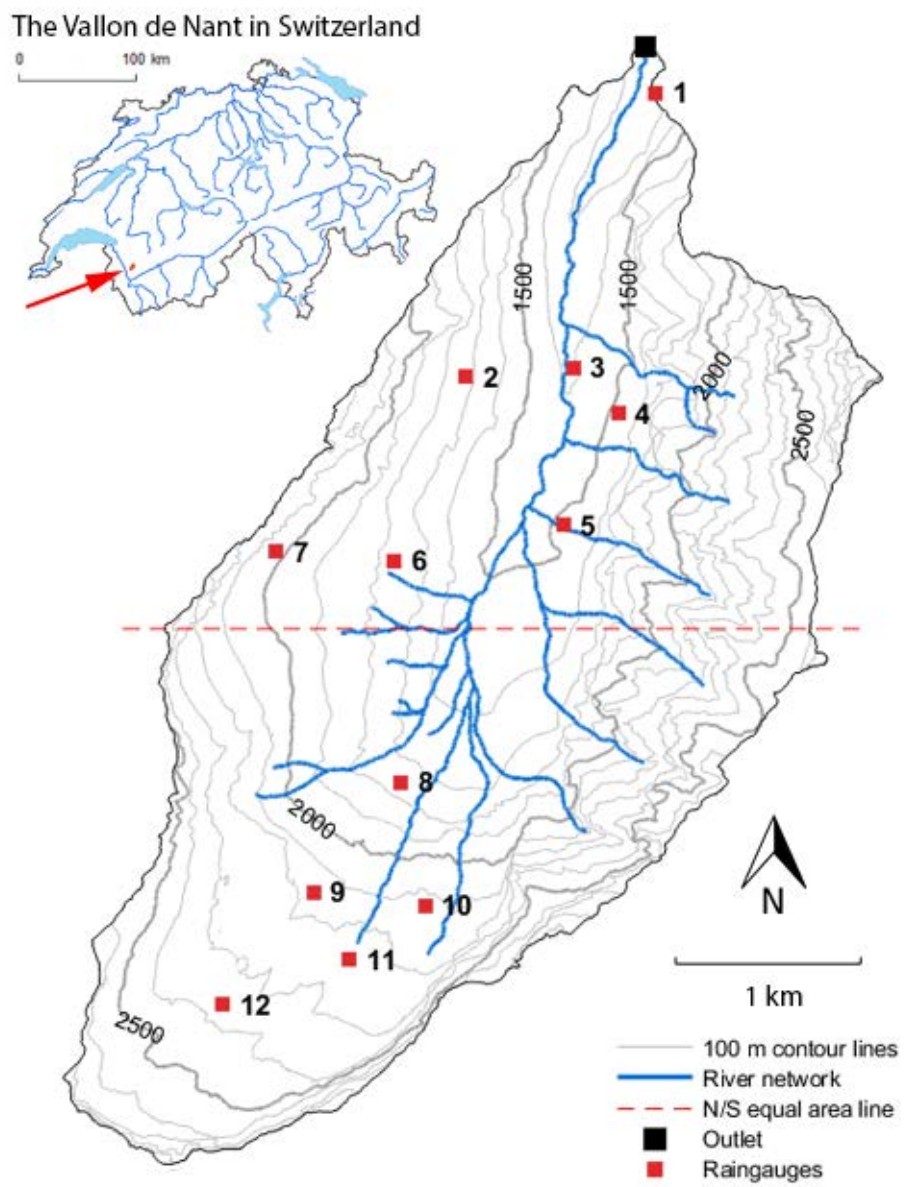

**Figure 1. Map of the Vallon de Nant and position of the 12 rain gauges. The discharge is measured on the main river at the outlet (46.25301 N / 7.10954 E in WGS84 coordinates). The red dashed line splits the catchment area into two parts of equal area.**





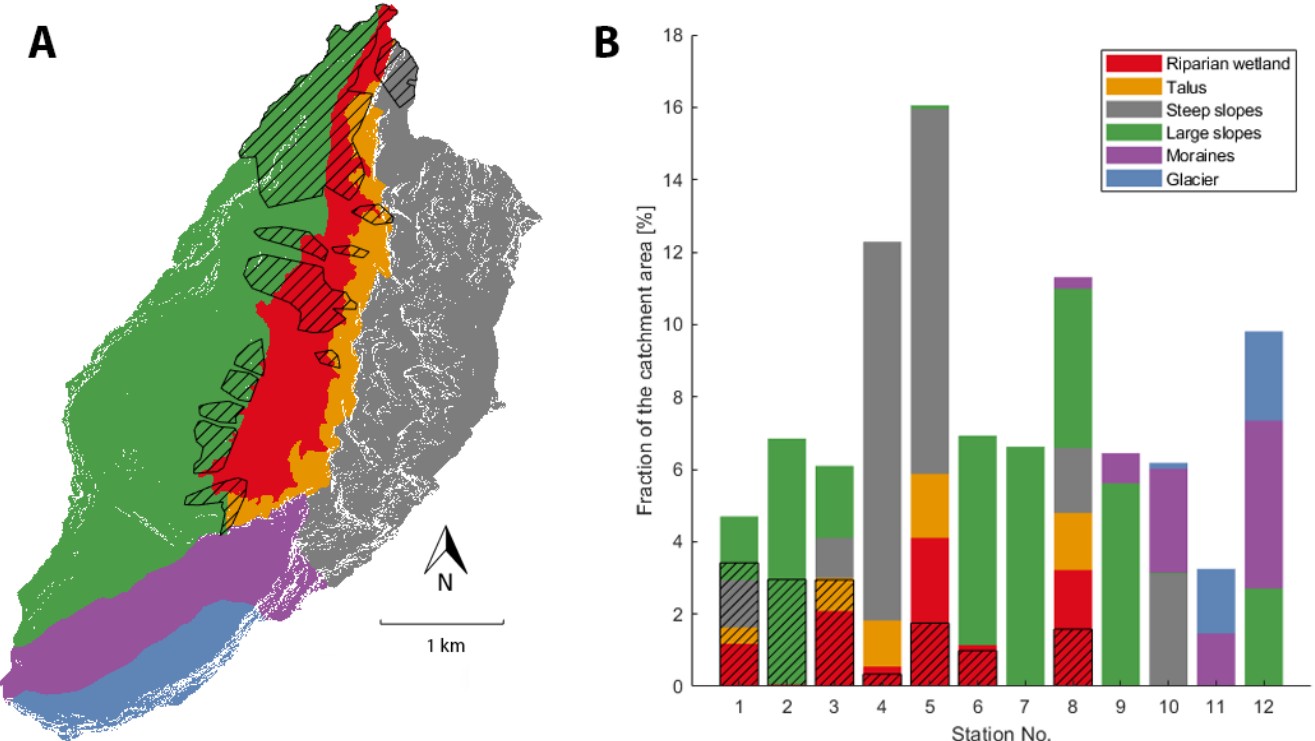

**Figure 2. Hydrological units in the Vallon de Nant (left) and their relative areas within the spatial imprint attributed to each rain**
**gauge (right). The hatched area corresponds to the densely forested area. For the spatial imprint of each station on the map, see**
**Figure 9.**






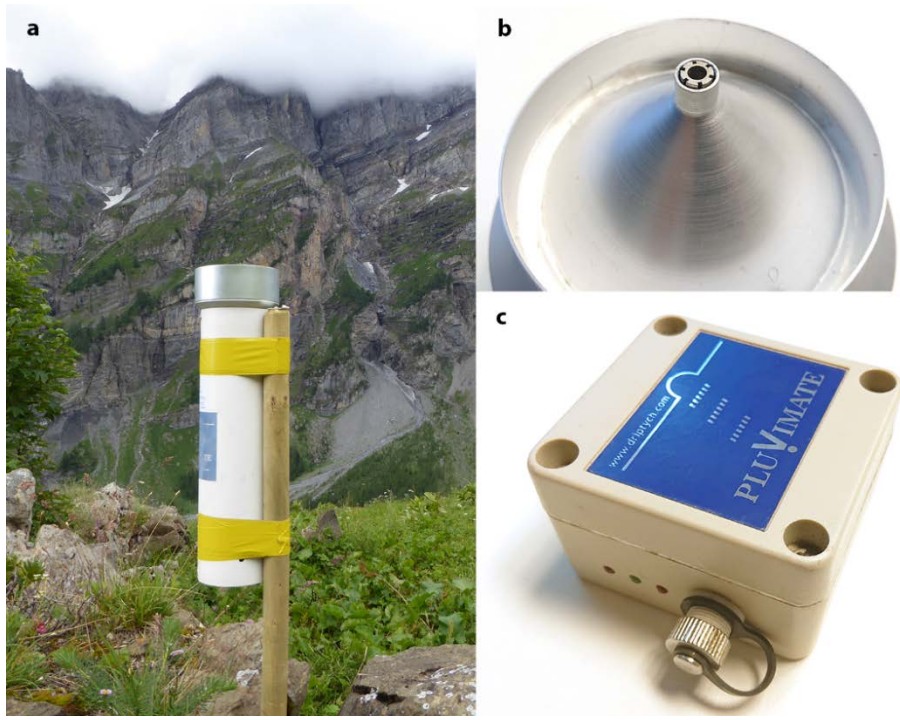

**Figure 3: Drop-counting rain gauge used for rainfall measures. The Pluvimate is set-up vertically between 0.8 and 1.2 meters above the ground level (a). A tip at the end of the funnel (b) create a calibrated drop of water that falls on the sensor, (c) which counts and records the number of drops during a given amount of time.**

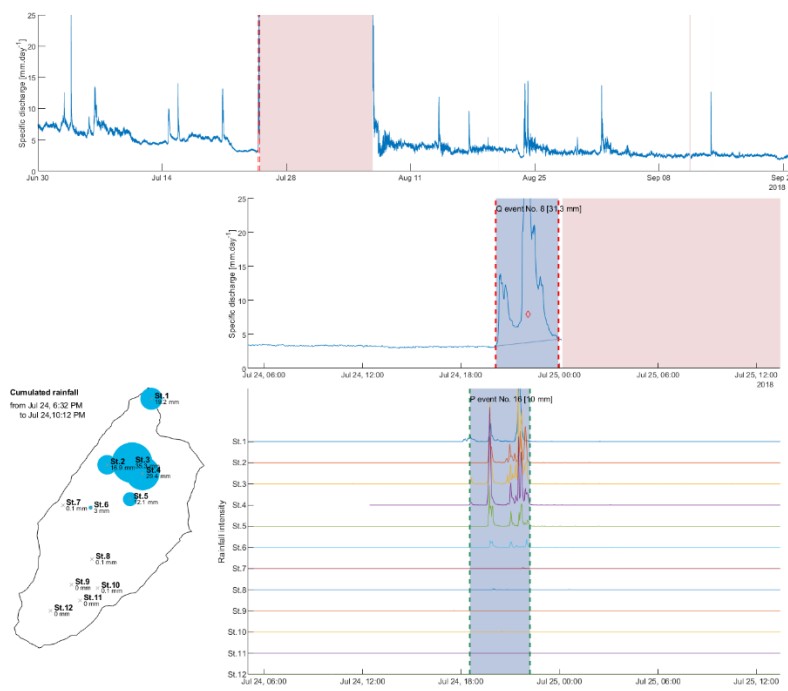

**Figure 4 Summary of the recorded rainfall and discharge for the rainfall event of July 24th 2018 at 6:32 PM (UTC)**


**Figure 5. Summary of the recorded rainfall and discharge for the rainfall events of August 24th 2018 at 2:46 AM (top) and August 29th 2018 at 11:52 AM (bottom).**







**Figure 6. Scatter plots of rainfall and streamflow response metrics. In A, in addition to the dashed 1/1 line, the dotted 1/2 and 2/1 lines represent the limit beyond which the rain field is considered asymmetrical (below $I_{ASYM}$=-0.33 and above $I_{ASYM}$=0.33). For the plots C, D, E and F, the rainfall events (P events) without river reaction (Q event) are not included in the linear regressions and the r² computations. The outlier event of July 24[th] (discussed in 4.1) is flagged by the letter $a$ on A, C and D, but is not visible on G.**






**Table 1. List and definition of the metrics used in this study, with corresponding parameter name or abbreviation used in the text.**

| Description | Notation, Unit |
| --- | --- |
| Rainfall interpolated over entire catchment | $P_{ALL}$, mm |
| Rainfall interpolated over north half (lower part) of catchment | $P_{LOWER}$, mm |
| Rainfall interpolated over south half (upper part) of catchment | $P_{UPPER}$, mm |
| Index of spatial asymmetry of rainfall between upper and lower parts | $I_{ASYM}$, - |
| Distance of rainfall spatial center of mass to stream network (along hillslopes) | $D_{HILLS}$, m |
| Distance of rainfall spatial center of mass to outlet along the stream network | $D_{STREAM}$, m |
| Fast streamflow rainfall | $Q_{FAST}$, mm |
| Rainfall runoff coefficient | RC, - |
| Lag time between rainfall event start and river reaction start | $P_{START}\_Q_{START}$, min |
| Cumulated amount of rainfall for the last X days | Wet. $X$ days, mm |







**Table 2. List of the tested predictors for the RC pure quadratic regression, and their corresponding statistics: root mean square error (RMSE), coefficient of determination (R²), variance of residuals (var. residuals), p-value, Akaike criterion (AIC), AIC ranking, corrected Akaike criterion (AICc) and AICc ranking. The acceptable p-values (<0.05) and first 3 ranks are highlighted.**

| Predictor 1 | Predictor 2 | RMSE | $R^2$ | var. residuals | p-value | AIC | rank AIC | AICc | rank AICc |
|---|---|---|---|---|---|---|---|---|---|
| $P_{LOWER}$ | | 0.38 | 0.01 | 0.12 | 0.92 | -26.5 | 22 | -24.3 | 13 |
| $P_{UPPER}$ | | 0.33 | 0.26 | 0.09 | 0.17 | -30.7 | 7 | -28.5 | 4 |
| $P_{ALL}$ | | 0.36 | 0.10 | 0.11 | 0.52 | -27.9 | 15 | -25.7 | 11 |
| $I_{ASYM}$ | | 0.34 | 0.19 | 0.10 | 0.27 | -29.5 | 10 | -27.3 | 7 |
| $D_{HILLS}$ | | 0.35 | 0.18 | 0.10 | 0.31 | -29.2 | 13 | -27.0 | 8 |
| Wet. 1 day | | 0.36 | 0.11 | 0.11 | 0.49 | -28.0 | 14 | -25.8 | 10 |
| Wet. 2 days | | 0.33 | 0.23 | 0.10 | 0.21 | -30.2 | 9 | -28.0 | 6 |
| Wet. 3 days | | 0.27 | 0.50 | 0.06 | 0.02 | -36.5 | **2** | -34.3 | **1** |
| Wet. 4 days | | 0.30 | 0.36 | 0.08 | 0.07 | -33.0 | 5 | -30.8 | **2** |
| $P_{LOWER}$ | $I_{ASYM}$ | 0.36 | 0.26 | 0.09 | 0.50 | -26.8 | 19 | -20.2 | 20 |
| $P_{UPPER}$ | $I_{ASYM}$ | 0.35 | 0.31 | 0.09 | 0.40 | -27.8 | 16 | -21.1 | 17 |
| $P_{ALL}$ | $I_{ASYM}$ | 0.36 | 0.27 | 0.09 | 0.49 | -27.0 | 18 | -20.3 | 19 |
| $D_{HILLS}$ | $I_{ASYM}$ | 0.36 | 0.24 | 0.09 | 0.55 | -26.4 | 23 | -19.7 | 23 |
| Wet. 1 day | $I_{ASYM}$ | 0.36 | 0.28 | 0.09 | 0.47 | -27.1 | 17 | -20.5 | 18 |
| Wet. 2 days | $I_{ASYM}$ | 0.32 | 0.43 | 0.07 | 0.20 | -30.6 | 8 | -23.9 | 14 |
| Wet. 3 days | $I_{ASYM}$ | 0.25 | 0.63 | 0.05 | 0.03 | -37.1 | **1** | -30.4 | **3** |
| Wet. 4 days | $I_{ASYM}$ | 0.29 | 0.52 | 0.06 | 0.09 | -33.4 | 4 | -26.7 | 9 |
| $P_{LOWER}$ | $D_{HILLS}$ | 0.37 | 0.21 | 0.10 | 0.64 | -25.7 | 24 | -19.0 | 24 |
| $P_{UPPER}$ | $D_{HILLS}$ | 0.33 | 0.38 | 0.08 | 0.27 | -29.3 | 12 | -22.7 | 16 |
| $P_{ALL}$ | $D_{HILLS}$ | 0.36 | 0.25 | 0.09 | 0.54 | -26.5 | 20 | -19.8 | 21 |
| Wet. 1 day | $D_{HILLS}$ | 0.36 | 0.25 | 0.09 | 0.54 | -26.5 | 21 | -19.8 | 22 |
| Wet. 2 days | $D_{HILLS}$ | 0.33 | 0.38 | 0.08 | 0.26 | -29.5 | 11 | -22.8 | 15 |
| Wet. 3 days | $D_{HILLS}$ | 0.27 | 0.57 | 0.05 | 0.06 | -34.9 | **3** | -28.3 | 5 |
| Wet. 4 days | $D_{HILLS}$ | 0.30 | 0.48 | 0.06 | 0.13 | -32.1 | 6 | -25.4 | 12 |






**Figure 7. Best network for 1 to 5 stations resulting from the minimization of RMSE for the $P_{ALL}$ (23 events), the $I_{ASYM}$ (23 events) and the RC (6 events). The red dashed line splits the catchments into two parts of equal area.**



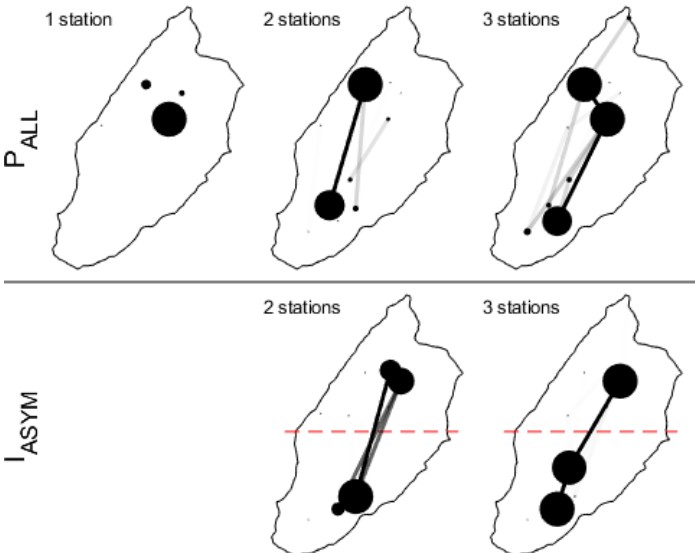

**Figure 8. Sensitivity test over the best network from 1 to 3 station, evaluated by removing from 1 to 3 events over the 23 events (2047 combinations) for the $P_{ALL}$ and $I_{ASYM}$. The result is presented graphically: larger dots and wider links represent configurations that are found more frequently than others over the different altered simulations. The red dashed line splits the catchments into two parts of equal area.**

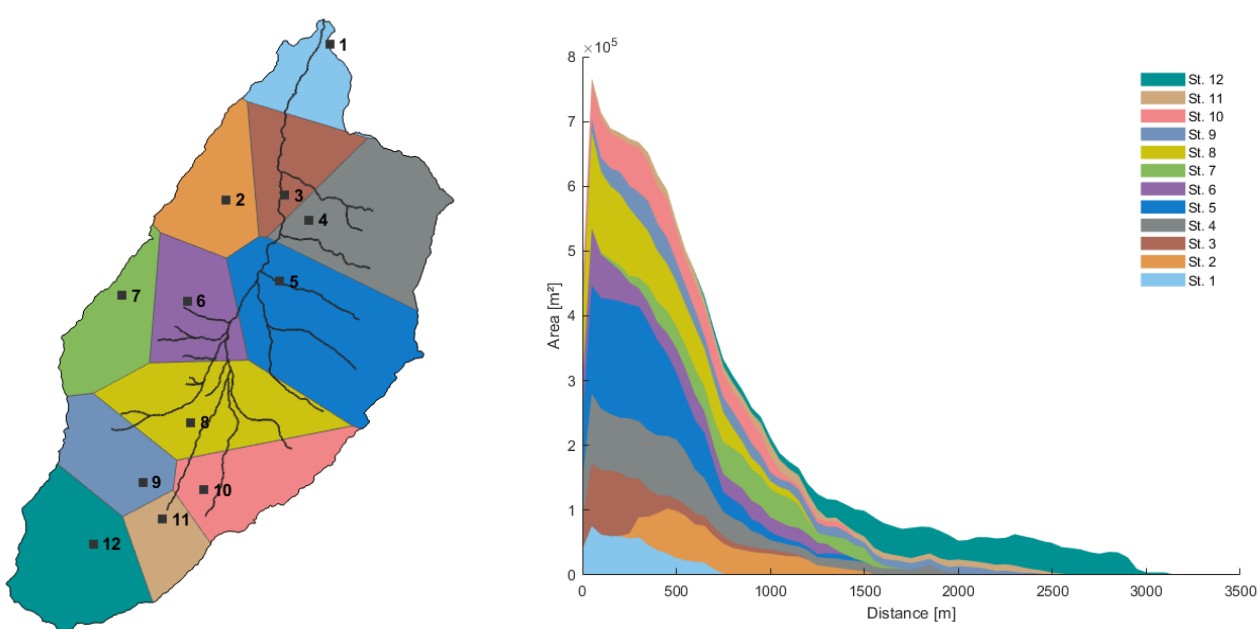

**Figure 9. Cumulative distribution of distances to the work for the spatial imprint of each weather station.**