# Peer review of "On the value of high density rain gauge observations for small Alpine headwater catchments"

_Hydrology and Earth System Sciences, 2019_

## Referee Comment (RC1) · Anonymous Referee #1 · 14 Feb 2020

The paper aims at highlighting the values of high density rain gauges networks for hydrological purposes in small catchment of mountainous areas. The topic is interesting and relevant for the community. It furthermore has other potential applications in urban areas which are also small and quickly reactive catchments where rainfall variability has strong consequences.

Although quite short (and it should be stressed more clearly that it is a limitation of the study), the data set is relevant. The paper is well presented and easy to read (except for Fig. 8 and corresponding comment, see below).

However, I think that the indicators used to characterize the rainfall variability are too simplistic (basically an asymmetry indicator splitting the catchment in two) to enable robust conclusion. The indicators of hydrological behaviour also seem quite simplistic.

[Figure]

And this is confirmed by the low scores and quality of regressions that are found. I believe that indicators enabling to grasp more precisely rainfall variability and its consequences should be used. I guess that this would enable to highlight more precisely the importance of dense networks of rainfall measurement devices.

Detailed comments:

- l.15 (abstract) : "the identification of key hydro-meteorological metrics that explain the runoff coefficient and lag times(e.g. total event rainfall, center of mass of the precipitation field)" : depending on the application there could be other indicators as well.

- Introduction : I believe it would be worth mentioning urban applications. Indeed, there have recently been numerous papers highlighting the need for high resolution rainfall data for these small catchments.

- l. 155-157 : I do not see where is the "steel sponge" on Fig. 3. Could you please highlight it ? It might be interesting to test the sensitivity of the results to this issue.

- Eq. 1 : it seems to be a very simplistic indicator of the rainfall variability. Many other have been developed to characterize much better the rainfall variability.

- Eq. 2 : given the fractal nature of river networks, how the river network was determined ? i.e. at which resolution was the upstream network not taken into account ?

- l. 230-231 : please clarify of the fast runoff is computed.

- l 237 : I guess it should be a reference to Table 2.

- Section 3.5 : I am not sure that AIC is needed, if the "corrected" version is also used.

- Section 4.4 : basically the absence of good models seems to suggest that the indicator used are too simplistic and do not enable to grasp the hydrological behaviour.

- Fig. 8 : I found it difficult to understand what is done. Could you please clarify ?

[Figure]

---

## Referee Comment (RC2) · Anonymous Referee #2 · 20 Feb 2020

I was a priori very interested by this work and I found the introduction of the article well focussed and documented. I was a bit sceptic however about the first objective of characterizing "the effect of spatial location of rainfall fields on the timing and amplitude of the hydrological response", based on data analysis only (no modelling) and for such a small watershed. I began to be disillusioned:

in page 6, with (i) the lack of analysis of the spatial variability of rainfall and, e.g. with the implementation of the Thiessen's method for 2-min rain resolution data; (ii) the fact that Figs. 4 and 5 are hardly readable; (iii) the too many references to the supplementary material, starting on line 181 (the general reader will not follow you there ; the article has to be synthetic and "self-contained");

in page 7, with the choice of the spatial rainfall asymmetry index. The shape of the

watershed matters, so why not consider differences in distance and amplitude between the catchment and the rainfall "width functions", as proposed by several authors in the literature; the topography could be included as well in some way, a metrics to be invented, which would be relevant especially in such a high-mountain context;

in page 8, with consideration of the initial wetness conditions as "hydrological response metrics" (while this variable is more on the forcing side), the absence of standard indices on lag times between the hyetogram and the hydrogram (e.g. response time, time of concentration, etc), the way you have determined the runoff volume. Among other points, it is indeed difficult to get an idea of the response time of the watershed, which could drive a basic discussion about time and spatial sampling issues, (e.g. Berne et al., J. Hydrol., 2004, 299, 166-179);

in page 8 with the description of the statistical analysis (pure quadratric regression) while Fig. 6 is based on simple linear regression and the regression attempts presented in Table 2 could have been done with standard multiple regression. Note that, rather than p-values and AIC criteria listed in Table 2, the number of points considered in each regression would be sufficient for the reader to assess the robustness of the inferences. (But more importantly, I doubt that any statistical technique of forcing and hydrological response variables will be able to replace a hydrological model . . .)

In addition, Fig. 6a could have closed rapidly the debate on the spatial variability of rainfall at the scale of this watershed. Heterogeneous events, with significant rainfall, occur once in a while and may impact the flood dynamics; but you do not give any evidence (and in my view there is no way to get it without a model) of this impact in the article.

The cases with runoff coefficients greater than 1 are interesting, especially the July 24th case. Indeed, the rainfall sampling in the steepest part of the watershed is probably deficient and it will be hard to obtain it with raingauges. Is there any hope to integrate some information from the Swiss radar network to compensate for this lack of data in

this area, and eventually over the entire watershed?

Therefore, although I recognize that there is a huge field work done, I think the analyses and presentation of this study require some additional effort before the article can be accepted for publication.

With respect to the state of the art presented in the introduction, I may recommend the authors to read (and eventually to refer to) two articles by I. Emmanuel et al. in J. Hydrol. 2015 (531, 337-348) and 2017 (555, 314-322) (which I did not co-authored, I swear!).

---

## Short Comment (SC1) · 12 Mar 2020

We would like to thank the reviewer for the overall positive assessment of our manuscript and the insightful propositions of improvement. We provide hereafter a detailed response on how we plan to revise our manuscript for each comment. The original comments are in italic, our response in normal font. For cross-referencing, we numbered the comments.

**Overall comments**

*1) The paper aims at highlighting the values of high density rain gauges networks for hydrological purposes in small catchment of mountainous areas. The topic is interest-*

[Figure]

*ing and relevant for the community. It furthermore has other potential applications in urban areas which are also small and quickly reactive catchments where rainfall variability has strong consequences. Although quite short (and it should be stressed more clearly that it is a limitation of the study), the data set is relevant.*

We will stress this limitation of the study in the revised version.

*2) The paper is well presented and easy to read (except for Fig. 8 and corresponding comment).*

Thanks for the positive assessment, we will improve Fig. 8 (please see below the answer to the point No. 13).

*3) However, I think that the indicators used to characterize the rainfall variability are too simplistic (basically an asymmetry indicator splitting the catchment in two) to enable robust conclusion. The indicators of hydrological behavior also seem quite simplistic. And this is confirmed by the low scores and quality of regressions that are found. I believe that indicators enabling to grasp more precisely rainfall variability and its consequences should be used. I guess that this would enable to highlight more precisely the importance of dense networks of rainfall measurement devices.*

The presented indicators can indeed seem simplistic, but we would like to underline that this choice was motivated by the observed spatial rainfall structures and the catchment shape. This led to indicators that evaluate the spatial heterogeneity of precipitation from two perspectives: (a) independently from precipitation location (asymmetry index), and (b) in relation with rainfall location relatively to the hydrological network ($D_{HILLS}$ and $D_{STREAM}$). To improve the set of precipitation indicators we propose to add i) a new indicator of rainfall variability that characterizes the spatial heterogeneity of precipitations (see detailed comment No. 7 hereafter), and ii) to use "width functions" in our analysis as suggested by the referee No. 2.

**Detailed comments:**

*4) l.15 (abstract): "the identification of key hydro-meteorological metrics that explain the runoff coefficient and lag times (e.g. total event rainfall, center of mass of the precipitation field)": depending on the application there could be other indicators as well.*

We will reformulate the corresponding sentence to make clear that these are the indicators that we used.

*5) Introduction: I believe it would be worth mentioning urban applications. Indeed, there have recently been numerous papers highlighting the need for high resolution rainfall data for these small catchments.*

Thanks for pointing this omission out. We did not mention urban applications because we made the assumption that the need for high resolution data in urban applications is much more evident than for rural catchments. We will mention urban applications explicitly in the revised version.

*6) l. 155-157: I do not see where is the "steel sponge" on Fig. 3. Could you please highlight it? It might be interesting to test the sensitivity of the results to this issue.*

The steel sponge is indeed not visible on the pictures. We will add/pick a picture showing the sponge in place in the revised version. We propose first to verify the effect of the sponge experimentally (using the same experimental setup as the Pluvimate calibration detailed in Appendix A) to evaluate i) the delay caused by the sponge on low rainfall intensities and ii) the amount and timing of the delayed rainfall after the actual end of the rainfall event. If the results show a significant effect, in a second step we will evaluate the significance of theses artefacts on metrics used in this study.

*7) Eq. 1: it seems to be a very simplistic indicator of the rainfall variability. Many other have been developed to characterize much better the rainfall variability.*

We agree that an indicator for intra-event rainfall variability is missing. We propose to

use this indicator presented in Smith et al. (2004):

$$\sigma_t = \sqrt{\frac{\sum_{i=1}^{N} P_i^2}{N} - \frac{\left(\sum_{i=1}^{N} P_i\right)^2}{N^2}} \qquad (1)$$

$$I_\sigma = \frac{\sum \sigma_t P_t}{\sum P_t} \qquad (2)$$

with $P_i$ the rainfall amount at the station $i$ at the time step $t$ and $N$ the number of stations.

*8) Eq. 2: given the fractal nature of river networks, how the river network was determined? i.e. at which resolution was the upstream network not taken into account?*

The accuracy of the river network extent presented here is quite exhaustive: streams in the Vallon de Nant appear punctually at well-defined springs; accordingly, we identified the stream channel heads based on these springs (identified in the field). Exceptions are on the east side of the catchment, which is hardly accessible, where the stream channel heads are identified from topographic maps. These details about the river network identification will be added to the revised version.

*9) l. 230-231: please clarify of the fast runoff is computed.*

For reasons detailed in Section 3.3 of the original manuscript, streamflow events are identified manually from the hydrograph. Thus, an event starts at $t_{START}$ with a discharge $Q_{START}$ and ends at $t_{END}$ with a discharge $Q_{END}$. A straight line is then drawn between these two coordinates. The integration of the area above this straight line (the area between the straight line and the discharge curve) corresponds to what we consider as the amount of fast runoff during the event. The description of runoff computation will be clarified in the revised version.

[Figure]

*10) l 237: I guess it should be a reference to Table 2.*

Thanks, it will be corrected.

*11) Section 3.5: I am not sure that AIC is needed, if the "corrected" version is also used.*

We agree, we will use only the corrected version AICc.

*12) Section 4.4: basically, the absence of good models seems to suggest that the indicator used are too simplistic and do not enable to grasp the hydrological behavior.*

In our view, the absence of a good model indicates that there is no simple linear relationship between the observed variables and the runoff response. At the same time, the limited number of observed events prevents the use of a more complex model, which is a classical problem in comparable hydrological studies. The revised version will use an additional rainfall indicator. We will furthermore extend the statistical analysis (use multiple regression) and discuss in more detail the results.

*13) Fig. 8: I found it difficult to understand what is done. Could you please clarify?*

The aim of the sensitivity test presented in Fig. 8 is to evaluate the robustness of best weather stations network results presented in Fig. 7. We performed the same estimation, but removed 1, 2 or 3 out of the 23 events from the data set. Fig. 8 summarizes, for each possible dataset (testing all possible combinations of event), how often a given observational network was chosen as the best network. The more often a given station is retained as being part of the optimal network, the larger the symbol in the figure. In addition, the figure represents how often two stations are part of the same optimal network. The more often two stations are retained together, the wider the line between them. As Fig. 8 shows, for networks of 2 stations, the 2 same stations are the best choice most of the time (with the same holds for a 3-stations network).

REFERENCES:

Michael B. Smith, Victor I. Koren, Ziya Zhang, Seann M. Reed, Jeng-J. Pan, Fekadu Moreda, Runoff response to spatial variability in precipitation: an analysis of observed data, Journal of Hydrology, Volume 298, Issues 1–4, 2004, Pages 267-286, ISSN 0022-1694, https://doi.org/10.1016/j.jhydrol.2004.03.039.

---

## Author Comment (AC2) · 13 Mar 2020

Answers to referee #2

We would like to thank the reviewer for the detailed comments about our manuscript, and the insightful propositions of improvement. We provide hereafter a detailed response on how we plan to revise our manuscript for each comment. The original comments are in italic, our response in normal font. For cross-referencing, we numbered the comments.

*1) I was a priori very interested by this work and I found the introduction of the article well focussed and documented. I was a bit sceptic however about the first objective of characterizing "the effect of spatial location of rainfall fields on the timing and amplitude of the hydrological response", based on data analysis only (no modelling) and for such a small watershed. I began to be disillusioned:*

We will make sure the revised manuscript version does not raise too high expectations in the abstract.

*2) in page 6, with (i) the lack of analysis of the spatial variability of rainfall and, e.g. with the implementation of the Thiessen's method for 2-min rain resolution data;*

It is true that using a basic interpolation approach for 2-min resolution data does not fully take advantage of the quality of the rain dataset. Therefore, we propose to replace the Thiessen interpolation method by a high-resolution interpolation procedure developed by Benoit et al. (2018). In a nutshell, it aims at generating an ensemble of stochastic space-time rain fields constrained by the actual observations at raingauge locations (over 20 realizations), and to use this ensemble to interpolate sparse rain observations.

As an illustration, the Fig 1 shows 2 examples of rainfall events interpolated from the Thiessen method and the method from Benoit et al. (2018). The plots for all the 48 events will be uploaded later in the discussion. The total rainfall amounts computed with both methods is summarized in the Table 1.

[Figure]

*Figure 1. Example for 2 events of rainfall amounts interpolated over the catchment using the Thiessen method and the method developed by Benoit et al. (2018).*

**Table 1.** Summary of total rainfall amounts computed for each rainfall event, using the Thiessen interpolation method, and the stochastic approach developed by Benoit et al. (2018).

| P event No. | Thiessen method Total P [mm] | Benoit et al. (2018) Total P [mm] | Benoit et al. (2018) std(P) [mm] |
|---|---|---|---|
| 1 | 3,2 | 2,9 | 0,25 |
| 2 | 8,0 | 7,7 | 1,02 |
| 3 | 1,3 | 1,2 | 0,13 |
| 4 | 13,4 | 12,1 | 2,06 |
| 5 | 1,2 | 1,2 | 0,02 |
| 6 | 1,7 | 1,6 | 0,13 |
| 7 | 8,1 | 8,2 | 0,29 |
| 8 | 1,5 | 1,5 | 0,05 |
| 9 | 20,8 | 20,2 | 0,49 |
| 10 | 19,2 | 18,7 | 0,95 |
| 11 | 11,5 | 10,7 | 0,68 |
| 12 | 3,2 | 3,0 | 0,30 |
| 13 | 18,5 | 18,8 | 0,28 |
| 14 | 1,3 | 1,1 | 0,04 |
| 15 | 1,7 | 1,6 | 0,11 |
| 16 | 10,6 | 8,0 | 1,33 |
| 17 | 4,0 | 4,3 | 0,56 |
| 18 | 6,9 | 6,8 | 0,24 |
| 19 | 5,1 | 5,0 | 0,39 |
| 20 | 1,3 | 1,2 | 0,10 |
| 21 | 1,2 | 0,9 | 0,26 |
| 22 | 42,4 | 43,5 | 2,58 |
| 23 | 2,4 | 1,9 | 0,57 |
| 24 | 2,4 | 2,3 | 0,16 |
| 25 | 1,5 | 1,5 | 0,03 |
| 26 | 3,7 | 3,7 | 0,07 |
| 27 | 2,9 | 2,9 | 0,34 |
| 28 | 2,7 | 2,9 | 0,13 |
| 29 | 11,2 | 11,1 | 0,67 |
| 30 | 1,6 | 1,5 | 0,09 |
| 31 | 11,5 | 11,9 | 1,02 |
| 32 | 2,3 | 2,3 | 0,32 |
| 33 | 3,2 | 3,2 | 0,25 |
| 34 | 23,6 | 22,1 | 0,59 |
| 35 | 8,5 | 8,1 | 0,13 |
| 36 | 1,2 | 1,2 | 0,06 |
| 37 | 3,1 | 3,0 | 0,26 |
| 38 | 2,6 | 2,6 | 0,24 |
| 39 | 1,0 | 0,8 | 0,13 |
| 40 | 8,4 | 7,8 | 0,43 |
| 41 | 5,8 | 4,8 | 0,25 |
| 42 | 3,4 | 3,4 | 0,21 |
| 43 | 10,7 | 11,4 | 0,32 |
| 44 | 4,4 | 4,2 | 0,13 |
| 45 | 12,3 | 10,9 | 0,40 |
| 46 | 1,8 | 1,8 | 0,04 |
| 47 | 2,5 | 2,4 | 0,21 |
| 48 | 8,2 | 8,0 | 0,20 |

*3) (ii) the fact that Figs. 4 and 5 are hardly readable;*

The figures 4 and 5 of the paper will be simplified in order to make them more readable. In particular, we will delete the top plot showing the location of the event within the whole period of observation, which is not essential. The 3 remaining plots per figure (hydrogram, hyetograms and the map) will be easier to read this way.

*4) (iii) the too many references to the supplementary material, starting on line 181 (the general reader will not follow you there; the article has to be synthetic and "self-contained");*

We agree and will considerably reduce the number of references to the supplementary material in the revised version.

*5) in page 7, with the choice of the spatial rainfall asymmetry index. The shape of the watershed matters, so why not consider differences in distance and amplitude between the catchment and the rainfall "width functions", as proposed by several authors in the literature; the topography could be included as well in some way, a metrics to be invented, which would be relevant especially in such a high-mountain context;*

The used distance metrics $D_{HILLS}$ and $D_{STREAM}$ are inspired by the catchment's width function: $D_{HILLS}$ and $D_{STREAM}$ also integrates the distance of precipitation, but differ from the width function as i) they are accounting separately for the travel along the river network and the travel distance to the network and ii) they correspond to an average value of the distances rather than to the entire distribution of distances as the width function.

We thank you for the suggestion to compare the catchment and the rainfall width functions directly. In the revised version, we plan to use one or several indicators that assess explicitly the differences between the width functions.

Some preliminary results are presented hereafter: Fig. 2 presents the river network in "wet" conditions and in "dry" conditions. These networks are used to calculate the width function of the Vallon de Nant in Fig. 3 using i) the distance to the outlet, ii) the distance to the river network in "wet" conditions and the iii) the distance to the river network in "dry" conditions. Fig. 4, 5 and 6 (at the end of the document) present the distribution of the precipitation amounts for the 48 events relatively to these 3 curves. We propose to use these metrics in addition to $D_{HILLS}$ and $D_{STREAM}$ in the revised version of the paper.

[Figure]

*Figure 2. Map of distances to the river network in 'wet' conditions (left) and 'dry' conditions (right)*

[Figure]

*Figure 3. Catchment width function using the distance to the outlet or to the river network in 'wet' and 'dry' conditions.*

*6) in page 8, with consideration of the initial wetness conditions as "hydrological response metrics" (while this variable is more on the forcing side), the absence of standard indices on lag times between the hyetogram and the hydrogram (e.g. response time, time of concentration, etc)*

These standard indices were calculated but not shown in the first version of the paper. They will be added and used in the revised version of the paper, with the aim of having one robust metric for the flow reaction in terms of quantity and one in terms of timing.

*7) [...] the way you have determined the runoff volume.*

The determination of runoff volume indeed relies on our expertise and implicates a possible estimation error. In response also to reviewer 1, the calculation will be explained in more details. We will also try to associate an uncertainty estimate to the runoff volume estimation: an uncertainty will be introduced for each start and end of a discharge event, giving each a low and high runoff volume. This uncertainty will propagate then into the rest of the analysis by using these 2 values.

*8) Among other points, it is indeed difficult to get an idea of the response time of the watershed, which could drive a basic discussion about time and spatial sampling issues, (e.g. Berne et al., J. Hydrol., 2004, 299, 166-179);*

The response time lag of the catchments (i.e. difference of start of rainfall events and start of streamflow reaction) varies between 12 and 90 minutes. We will explicitly include these estimates in the revised version and discuss the rainfall observation scheme (spatial and temporal resolution) with respect to this. The challenge hereby is to find relevant literature for rural, alpine catchments where surface runoff is not dominating (the key body of relevant literature is developed for surface runoff (Berne et al., 2004;Kim and Kim, 2018) or for hydrological models, as discussed in the introduction of the original manuscript (e.g.Huang et al., 2019).

*9) in page 8 with the description of the statistical analysis (pure quadratic regression) while Fig. 6 is based on simple linear regression and the regression attempts presented in Table 2 could have been done with standard multiple regression.*

We agree that the Fig. 6 is confusing as it mixes in the same figure the correlation among explanatory variables, and between explanatory variables and the response variable. The plots will therefore be separated into 2

figures in the revised version. In addition, we will extend the regression analysis for the revised version. Final results to be presented will depend on the effect of the additional rainfall indicators.

*10) Note that, rather than p-values and AIC criteria listed in Table 2, the number of points considered in each regression would be sufficient for the reader to assess the robustness of the inferences. (But more importantly, I doubt that any statistical technique of forcing and hydrological response variables will be able to replace a hydrological model…)*

We agree that AIC and the p-value could be removed. In contrast, the AICc will be kept to allow for model selection in case of different numbers of model parameters. We further comment on the need of using a hydrological model in our response to comment 11 below.

*11) In addition, Fig. 6a could have closed rapidly the debate on the spatial variability of rainfall at the scale of this watershed. Heterogeneous events, with significant rainfall, occur once in a while and may impact the flood dynamics; but you do not give any evidence (and in my view there is no way to get it without a model) of this impact in the article.*

Thank you for this comment. We agree that there is no other way to answer this question than using a hydrological model. We therefore propose to use a semi-lumped box model to assess the sensitivity of catchment to different rainfall patterns. This will be a semi-virtual experiment in the sense that we will calibrate the model parameters on the observed rainfall-streamflow dynamics once (to ensure a plausible parameterisation) but we will then keep the parameters constant to test the effect of different rainfall patterns on the rainfall response. This will be completed as follows i) rescaling the interpolated spatial rainfall fields to different total precipitation amounts and ii) assessing the variability of the discharge reactions in response to the various rescaled rainfall events.

We are aware that more advanced models, parametrizations and rain generation methods could be proposed to this problem, but we also aim to not turn this paper into a heavy modelling paper, which was not our first objective.

*12) The cases with runoff coefficients greater than 1 are interesting, especially the July 24th case. Indeed, the rainfall sampling in the steepest part of the watershed is probably deficient and it will be hard to obtain it with raingauges. Is there any hope to integrate some information from the Swiss radar network to compensate for this lack of data in this area, and eventually over the entire watershed?*

*MeteoSwiss* produces 1 x 1 km radar data with a 5 min time step, which is pretty coarse in comparison with the dimensions of the catchment (6 km x 5 km). In addition, Fig. 8 (hereafter) shows that the correlation between the Pluvimate and RADAR is rather low, due to both radar measurement errors and differences of support between point rain gauge data and spatially averaged radar images. Moreover, radar data over the Vallon de Nant catchment is especially affected by mountains around, which generates artefacts due to beam blocking. We will include this comparison (Fig. 8) in the supplementary material.

*13) With respect to the state of the art presented in the introduction, I may recommend the authors to read (and eventually to refer to) two articles by I. Emmanuel et al. in J. Hydrol. 2015 (531, 337-348) and 2017 (555, 314-322) (which I did not co-authored, I swear!).*

Thank you, we will add these references in the revised version of the paper.

*REFERENCES:*

Benoit, L., Allard, D., and Mariethoz, G.: Stochastic Rainfall Modeling at Sub-kilometer Scale, Water Resour Res, 54, 4108-475 4130, 10.1029/2018WR022817, 2018.

Berne, A., Delrieu, G., Creutin, J.-D., and Obled, C.: Temporal and spatial resolution of rainfall measurements required for urban hydrology, Journal of Hydrology, 299, 166-179, https://doi.org/10.1016/j.jhydrol.2004.08.002, 2004.

Foehn, A., García Hernández, J., Schaefli, B., De Cesare, G., Spatial interpolation of precipitation from multiple rain gauge networks and weather radar data for operational applications in Alpine catchments, Journal of Hydrology, Volume 563, 2018, Pages 1092-1110, ISSN 0022-1694, https://doi.org/10.1016/j.jhydrol.2018.05.027.

Huang, Y. C., Bardossy, A., and Zhang, K.: Sensitivity of hydrological models to temporal and spatial resolutions of rainfall data, Hydrology and Earth System Sciences, 23, 2647-2663, 10.5194/hess-23-2647-2019, 2019.

Kim, C., and Kim, D. H.: Effect of rainfall spatial distribution and duration on minimum spatial resolution of rainfall data for accurate surface runoff prediction, J. Hydro-environ. Res., 20, 1-8, 10.1016/j.jher.2018.04.001, 2018.

[Figure]

**Figure 4.** *Catchment width function w(x) and proportion of rainfall W$_P$(x) falling at a distance x [m] to the outlet for each event. The event ID is between brackets, followed by the total amount of rainfall during the event.*

[Figure]

**Figure 5.** *Catchment width function w(x) and proportion of rainfall W_P(x) falling at a distance x [m] to the river network in 'wet' conditions for each event. The event ID is between brackets, followed by the total amount of rainfall during the event.*

[Figure]

**Figure 6.** *Catchment width function w(x) and proportion of rainfall W$_P$(x) falling at a distance x [m] to the river network in 'dry' conditions for each event. The event ID is between brackets, followed by the total amount of rainfall during the event.*

[Figure]

**Figure 7.** *Distribution of C index calculated for 48 events using the width function up to the outlet and up to the river network in 'wet' and 'dry' conditions.*

[Figure]

*__Figure 8.__* Total amount of rainfall per event measured by the RADAR vs. the amount measured by the Pluvimate network (interpolated using the Thiessen method).

---

## Author Comment (AC3) · 13 Mar 2020

Please find in the supplement the figures showing for all 48 rainfall events of our dataset the results of the rainfall interpolation using both the method developed by Benoit et al. (2018) and the Thiessen method.

Anthony Michelon, Lionel Benoit, Harsh Beria, Natalie Ceperley and Bettina Schaefli.

Please also note the supplement to this comment: https://www.hydrol-earth-syst-sci-discuss.net/hess-2019-683/hess-2019-683-AC3-supplement.zip
* * *
683, 2020.

---

## Editor Comment (EC1) · Marie-Claire ten Veldhuis (Editor) · 6 Apr 2020

This paper presents a study of the effect spatial rainfall distribution on hydrologic response, based on a dense (1/km2) network of rainfall sensors in a small alpine catchment. The question of how spatial rainfall distribution affects hydrologic response has been addressed in many previous studies, several based on multiple catchments and longer series of rainfall events. The novelty of this study in its current set-up is therefore not convincing. Both reviewers point out that the indicators used to characterize rainfall variability are rather simplistic compared to what has been presented in previous studies. Furthermore, the results are not put in context of the literature (which should be done in the Discussion section), to show what new findings were achieved.
The reviewers have provided multiple suggestions for the use of more elaborate indica-

tors to develop the analysis, which are recognized in the authors response and some first examples are already shown. Given the amount of additional work that is foreseen, the results of which may change conclusions that were obtained, the work is likely to differ substantially from the original manuscript.

Based on these considerations and the assessment of the reviewers I encourage the authors to revisit the (valuable) dataset they have collected, to extend their methodology as discussed in the reviewer comments and author replies, in order to develop a substantially more in-depth analysis. Additionally, a more critical discussion in view of the existing literature for small-scale rainfall variability and hydrologic response (in small rural but also urban catchments, as pointed out by one of the reviewers) is warranted. The authors are welcome to submit the manuscript that will be developed based on the new outcomes as a new manuscript.

––––––––––––––––––––––––